# Evolution of nanopores in hexagonal boron nitride

Chunhui Dai[1,2,3], Derek Popple [2,3,4], Cong Su [1,2,3], Ji-Hoon Park [5], Kenji Watanabe [6], Takashi Taniguchi [6], Jing Kong [5] & Alex Zettl [1,2,3 ✉]

The engineering of atomically-precise nanopores in two-dimensional materials presents exciting opportunities for both fundamental science studies as well as applications in energy, DNA sequencing, and quantum information technologies. The exceptional chemical and thermal stability of hexagonal boron nitride (h-BN) suggest that exposed h-BN nanopores will retain their atomic structure even when subjected to extended periods of time in gas or liquid environments. Here we employ transmission electron microscopy to examine the time evolution of h-BN nanopores in vacuum and in air and find, even at room temperature, dramatic geometry changes due to atom motion and edge contamination adsorption, for timescales ranging from one hour to one week. The discovery of nanopore evolution contrasts with general expectations and has profound implications for nanopore applications of two-dimensional materials.

[1] Department of Physics, University of California at Berkeley, Berkeley, CA 94720, USA. [2] Materials Sciences Division, Lawrence Berkeley National Laboratory, Berkeley, CA 94720, USA. [3] Kavli Energy NanoSciences Institute at the University of California at Berkeley and the Lawrence Berkeley National Laboratory, Berkeley, CA 94720, USA. [4] Department of Chemistry, University of California at Berkeley, Berkeley, CA 94720, USA. [5] Department of Electrical Engineering and Computer Science, Massachusetts Institute of Technology, Cambridge, MA 02142, USA. [6] International Centre for Materials Nanoarchitectonics, National Institute for Materials Science, Tsukuba, Japan. ✉email: azettl@berkeley.edu

There is rapidly growing interest in local atomic-structure engineering of two-dimensional (2D) materials such as graphene[1,2], $MoS_2$[3,4], and hexagonal boron nitride (h-BN)[5–13]. Nanopores, ranging in size from just one missing atom to dozens or even hundreds of missing atoms, can have profound effects on the chemical[14], mechanical[15,16], and electro-optical properties of the material[17–19]. Nanopore technology could reshape various fields ranging from energy conversion and storage[20] to water purification[14] to quantum data technologies[15,17,21–24].

Of particular interest is h-BN, for which nanopores with highly reproducible shapes and sizes, with atomically precise edges, can be readily created[5–13]. Bulk h-BN is also especially chemically and thermally stable (even in an oxidizing environment)[25], suggesting highly stable nanopore configurations. Edge defects are found to effectively tune the band gap of h-BN[26], which enables the creation of highly reactive sites for hydrogen evolution reaction and facilitates the development of hydrogen-based green energy[20]. Channels formed in monolayer h-BN can serve as ultrasensitive solid-state nanopore sensors for DNA sequencing and advanced DNA-based information storage devices[27]. Vacancy-related defects in monolayer[17], multilayer[23], and interfacial[24] h-BN forms serve as controllable, efficient quantum emitters setting the stage for optically based quantum data transfer and computation technologies.

Various techniques have been employed to achieve nanopores in h-BN, including laser irradiation[28], focused ion beam (xenon[29,30], nitrogen[30], argon[30], and helium ions[31]), and condensed electron beam (e-beam)[5,7,11–13,15,32]. Among these techniques, condensed electron beam irradiation in a TEM or a scanning transmission electron microscope (STEM) stands out as the most precise and controllable tool due to its capability for selective knock-on energy transfer coupled with in situ atomic-scale imaging[12,13,32]. In h-BN, the nanopore shape (e.g., triangular, hexagonal, or otherwise) can be precisely tuned by selecting the temperature or irradiation dose during the electron irradiation process[5,7].

Many of the predicted nanopore properties for h-BN (and related 2D materials) depend sensitively on the pore size and edge geometry[12,33]. Therefore, it is critical to understand nanopore stability. Nanopore instability in graphene has been previously investigated[34,35]. However, because of the highly stable nature of bulk h-BN, it has generally been assumed that the atomic configuration of h-BN nanopores would be equally stable[21,22], especially under the relatively mild environmental conditions of the air at room temperature. In this report, we show that this conventional wisdom is wrong. We present an investigation of the evolution of nanopores in h-BN under different environmental conditions (vacuum and air, at room temperature). A condensed electron beam in a TEM system is used to create nanopores in mono- and multilayer h-BN. The nanopores are characterized by TEM immediately after being created, and exactly the same nanopores are again characterized via TEM after the sample is stored in a vacuum or air for selected time periods ranging from one hour to 1 week. TEM is used for this study because it has a shorter imaging acquisition time than STEM, and we thereby minimize undesirable imaging-induced nanopore changes. We find that nanopores in both monolayer and multilayer h-BN are highly stable in a vacuum but not in the air. Even at room temperature, the atomic configuration of h-BN nanopores evolves in the air spontaneously and dramatically within one hour. These findings suggest additional work on stabilizing the nanopores is required to employ them for reliable and stable devices.

## Results and discussion

Three types of h-BN are used in this study: monolayer h-BN, AA' stacked multilayer h-BN, and Bernal stacked (AB) multilayer h-BN. The monolayer and AB stacked multilayer h-BN are synthesized by a chemical vapor deposition (CVD) process as described in the methods and refs. [36,37]. As stated in our earlier work[37], the AB stacking order is achieved by employing low-pressure CVD on transition metals, iron or copper, and has been confirmed by selected area electron diffraction. A wet transfer process is used to transfer the membranes onto holey silicon nitride TEM grids[37] (see Supplementary Fig. S1). AA' stacked multilayer h-BN is directly exfoliated from crystals grown in Japan and dry transferred onto a TEM grid[38]. After transferring, the suspended membranes are imaged and irradiated by an 80 kV e-beam in a TEM system (JEOL 2010) to create the nanopores.

Figure 1a schematically shows the electron irradiation-induced nanopore formation process in suspended h-BN membranes. As reported in previous work[7,12,13,32], the nanopore preferably starts from a boron (B) vacancy, as B has a lower knock-on threshold compared to nitrogen (N). Then the adjacent N and B atoms are knocked off alternatively, forming a larger triangular nanopore terminated by nitrogen zigzag edges[5]. Figure 1b shows thus-formed triangular nanopores with various sizes in monolayer

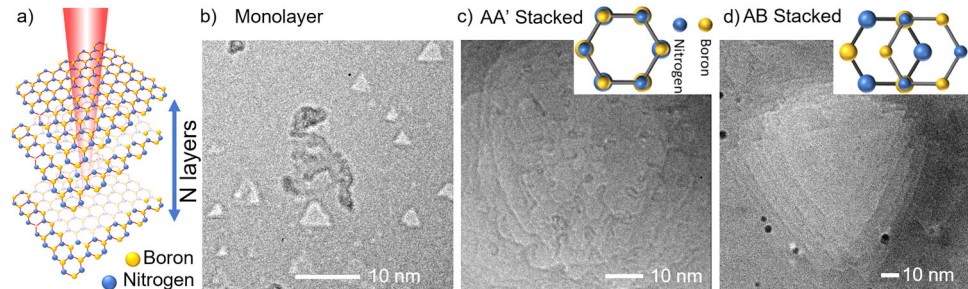

**Fig. 1 Schematics and TEM images showing the formation of nanopores in h-BN. a** Schematic of nanopore formation process in mono- or multi-layer h-BN using a condensed electron beam. A single boron vacancy is formed initially, and the nanopore grows into a larger nanopore. The electron beam is operated at 80 kV with a beam current of 40 A/cm². **b** TEM image of triangular nanopores formed in CVD-grown monolayer h-BN. **c** TEM image showing the formation of incomplete nanopores, i.e., pits and striping of layers in an exfoliated AA' stacked multilayer h-BN. The patterns shown in the TEM image are the edges of the nanopore pits. Due to the AA' stacking order, anti-directional triangular nanopores are formed in interlayers, leading to the loss of triangular geometry in the membrane. The membrane thickness is around ~10 nm. The inset shows schematically the AA' stacking configuration, where each atomic site has an N stacked over a B or a B stacked over an N. Upper layer atoms are drawn smaller for clarity. **d** TEM image of incomplete nanopores (pits) formed in Bernal (AB) stacked multilayer h-BN. The membrane thickness is around ~10 nm. The nanopores preserve the triangular geometry as in monolayer BN. The inset shows schematically the AB stacking configuration. As in the inset to **c**, upper layer atoms are drawn smaller for clarity.

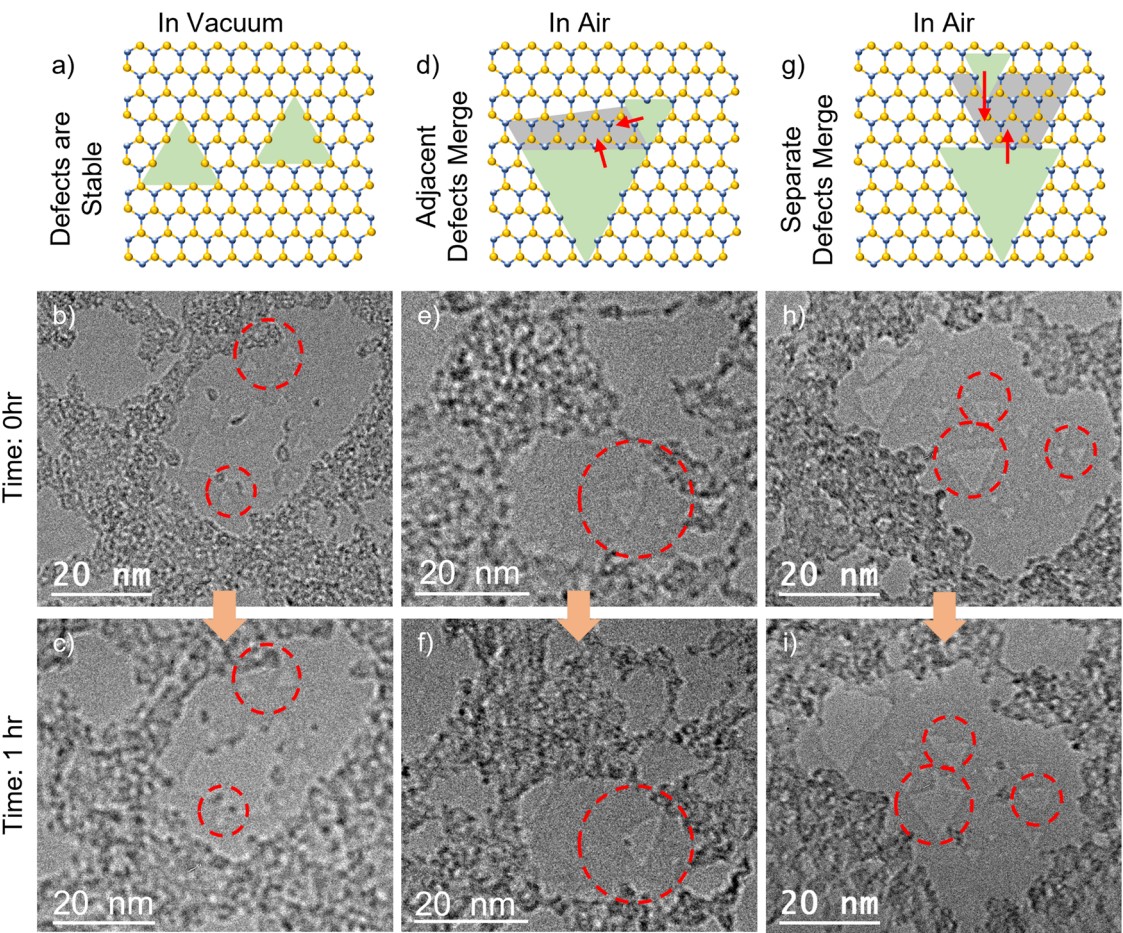

**Fig. 2 Evolution of nanopores in monolayer h-BN after being stored in vacuum or air for 1 h. a–c** are for vacuum and **d–i** are for air. **a, d, g** The schematics are generic representations of the geometries of the nanopores (and do not necessarily represent a scaled atomic mapping to the TEM images below). The green triangles represent the original geometry of the nanopores. **d, g** The red arrows show the merging direction of the nanopores, and the gray areas show the newly extended pore regime. **a** Schematics and **b, c** TEM images showing the defects **b** before and **c** after being stored in a vacuum for 1 h. **b** Initially, the sample supports suspended zigzag edges around large nanopores and smaller triangular nanopores (highlighted by the red circles). **c** The separate or adjacent distinct triangular nanopores are stable in a vacuum and preserve their original form. **d–i** Schematics and TEM images showing the deformation processes of **d–f** adjacent nanopores and **g–i** separate nanopores after being stored in air for 1 h. **e** TEM image of the two adjacent triangular nanopores (highlighted by the red circle). **f** The adjacent triangular nanopores are unstable in the air and merge into a larger triangular nanopore. **h** TEM image highlights three groups of nanopores (indicated by the red circles) with a separation from 1 to 5 nm. **i** The two separated nanopores have, upon exposure to air for 1 h, merged into a single nanopore, as shown schematically in (**g**). Orange arrows depict the flow of time.

h-BN. In multilayer BN, the electron beam may interact with adjacent layers, leading to more complex incomplete nanopores, i.e., pits[10]. In AA' stacked h-BN, all the B and N in the adjacent layers are alternately stacked together (Fig. 1c inset), losing the three-fold symmetry of the lattice[37]. As a result, the nanopore shape is not deterministic. Figure 1c shows incomplete nanopores formed on the surface of an AA' stacked h-BN. No preferred shape is observed. In contrast, for AB stacked h-BN, only half of the lattice sites are stacked in the form of B on N or N on B[37] (Fig. 1d inset), which allows the vacancies to preserve p3 symmetry as in the monolayer. Figure 1d shows incomplete nanopores formed on the surface of an AB-stacked h-BN. The triangular outlines can still be clearly identified. For the detailed study of nanopore evolution presented below, we use nanopores formed in monolayer and AB stacked multilayer h-BN, which have triangular geometry and allow us to clearly track geometry changes.

**Evolution of nanopores in monolayer hBN**. Figure 2 presents the nanopores' geometry change in monolayer h-BN 1 h after

their creation in a vacuum. Once the nanopores are created and imaged by an 80 kV TEM e-beam, the samples are withdrawn from the electron beam region and are stored at room temperature in a vacuum (Fig. 2a–c) or air with relative humidity ~40% (Fig. 2d–i). After 1 h, the same samples are returned to the vacuum and imaged to evaluate any spontaneous geometrical changes. To avoid unnecessary electron irradiation, a different area on the sample grid is first used for focusing. Once focused, the TEM viewing window is returned to precisely the same original nanopore region for secondary rapid structural characterization.

For nanopores that see only vacuum, the triangular nanopore remains intact, even for nanopores in close proximity to one other (examples are indicated by the red circles in Fig. 2b, c). For storage in air, both the adjacent nanopores (Fig. 2d–f) and the separated nanopores (Fig. 2g–i) are unstable, even at the minimum timescale of 1 h. The adjacent nanopores with a shared narrow edge (Fig. 2e, highlighted by the red circle) have merged, growing into an enlarged triangular nanopore (Fig. 2f, highlighted by the red circle). Separated nanopores are also unstable in the air. As shown in Fig. 2g, h, even though the large and small

## In Air

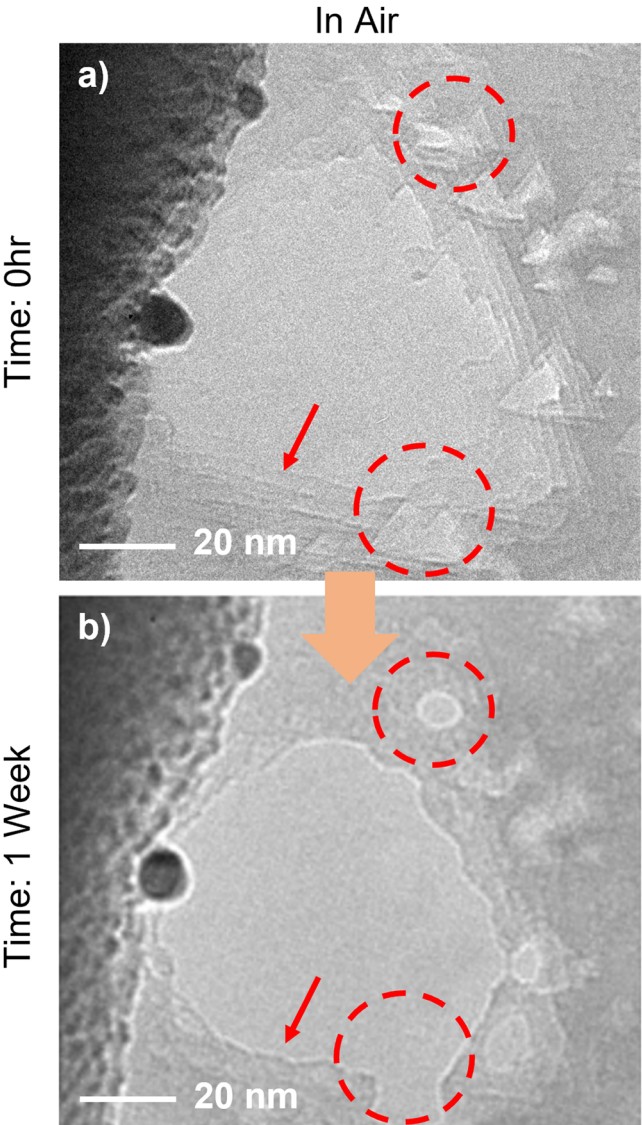

**Fig. 3 Longer-term evolution of nanopores in Bernal stacked multilayer h-BN stored in air. a** Triangular complete nanopores and incomplete nanopores (i.e., pits) are created using a condensed electron beam in multilayer h-BN. The electron beam is condensed to a 10–20 nm diameter with a beam current of ~40 A/cm$^2$. The overall nanopore formation process takes ~2 h. The thickness of the membrane is ~10 nm. The red arrow indicates the thin fringes near the large pores. The red circles highlight the pits partially etched area on the surface of the membrane. **b** After being stored in the air for one week, most of the triangular incomplete nanopores are unrecognizable. The sharp triangular edges near the pores are eroded, leading to a rounded nanopore. The red arrow indicates the eroded edges. Also, some incomplete nanopores are further etched into the h-BN membrane and form new nanopores, as highlighted by the red circles. The orange arrow indicates the flow of time.

triangular nanopores are separated by up to ~5 nm, a vacancy pathway is spontaneously formed between the two nanopores, making the two nanopores join into one (Fig. 2i). These observations indicate that the N-terminated edges of the nanopores are extremely unstable in the air. Unlike previously reported electron beam stimulated migration of such nanopores[7], the spontaneous deformation processes here observed are dominated by apparent etching/erosion of the atoms on the edge of the nanopores rather than the diffusion of atoms within the

lattice. This erosion-like process is conjectured to be caused by the reaction between the nanopore edge atoms with dangling bonds and the oxygen or other related gas molecules in the air.

**Evolution of nanopores in multilayer hBN.** To further examine the evolution of nanopores in h-BN, we conducted an experiment on AB-stacked multilayer h-BN at a longer time scale. Figure 3a shows the initial nanopores formed on a 10-nm thick AB-stacked h-BN membrane. The detailed nanopore formation processes are shown in supplementary Fig. S2. Both incomplete nanopores (i.e., nanopore pits that penetrate only some of the h-BN layers) and large through-pores can be observed. Also, the nanopores are all triangular in shape with clear edges. After being stored in the air for 1 week, these nanopores experience dramatic changes (Fig. 3b). First, almost no triangular incomplete nanopores remain at the same location. Second, the thinner membrane regions near the edges of the large pores are eroded (Fig. 3b, indicated by the red arrow). As a result, all the sharp edges of the triangular nanopores drilled through the membrane become rounded. Third, the incomplete nanopores can also erode normally to the surface, transforming a previously finite-depth pit into a round nanopore entirely piercing the membrane (Fig. 3b, highlighted by the red circles). These observations confirm a continuing longer-term evolution of the h-BN nanopore subjected to air.

**Surface contamination spreading.** Figure 4 presents another nanopore evolution phenomenon: that of surface contamination spreading. Figure 4a, c shows the initial nanopores with different sizes formed in the h-BN membranes. The dominant contaminants on the h-BN are presumably inevitable Poly(methyl methacrylate) (PMMA) residuals left from the wet transfer process[39]. But there this a clear boundary to a very clean regime that has no apparent contamination residuals (Fig. 4a, c, highlighted by the dashed white lines). Following initial nanopore creation and imaging, these two samples are stored in a vacuum or in the air for 15 h, respectively. In a vacuum, there is almost no change in the majority of the nanopores, and almost all of the originally clean area remains contamination free (a few spots of new contamination are highlighted by arrows in Fig. 4b). In contrast, the h-BN sample stored in air experiences significantly more change (Fig. 4d). When subjected to air, large portions of the originally clean area outlined in Fig. 4c become highly contaminated (Fig. 4d). The new contamination appears to be nucleated at edges of the existing patches of contamination, which apparently present preferential sites for molecule attachment. In the air more and more surface contaminants are built up, thereby "spreading" the surface contamination. This spreading can sometimes entirely obscure an original nanopore (Fig. 4d, indicated by the arrows). Overall, the air environment can significantly facilitate not only erosion but also contamination expansion processes and can quickly render, in a matter of hours, a well-engineered nanopore configuration in h-BN wholly ineffective.

### Conclusions

In conclusion, this work finds that different nanopores formed in mono- and multi-layer h-BN are stable in a vacuum but undergo dramatic changes in the air. In the air, nanopores can grow in size, which inevitably involves both boron and nitrogen removal at the pore edges. Existing nanopores can also be overrun by expanding surface contamination that primarily originates from adjacent patches of surface contamination. Notably, we do not observe any obvious changes to pristine regions (atomically perfect regions with no nanopores or contamination present) of

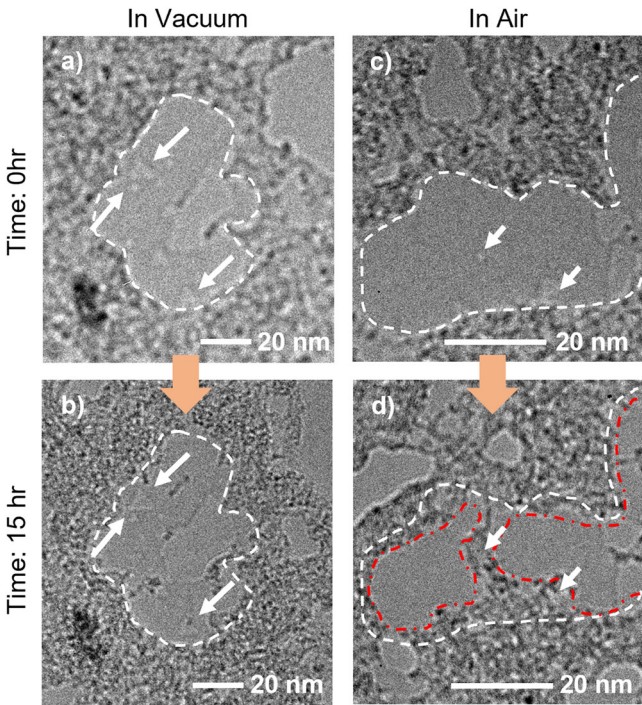

**Fig. 4 TEM images showing the relative stability of surface contamination in a vacuum and the spreading of surface contamination in the air for monolayer h-BN with nanopores. a, b** Sample always in a vacuum. The white dashed lines outline the boundary between the clean region (interior to the boundary) and the high surface contaminated region (exterior to the boundary). After 15 h in a vacuum, the boundary has not changed, and the nanopores present are largely unmolested (white arrows identify two representative nanopores). **c, d** Evolution of sample in air. The white dashed line in **c** outlines the initial boundary between clean (interior) and contaminated (exterior) regions; the same white dashed line is reproduced in (**d**). The red dashed dot line in d) shows the new contamination boundary after 15 h storage in air. The surface contamination has substantially progressed inward and has totally overrun the two nanopores identified with white arrows. The orange arrows depict the flow of time.

suspended monolayer or multilayer h-BN, in a vacuum or in an air environment. This is key to the excellent stability of bulk h-BN. We also surmise that h-BN nanopores covered by at least one monolayer of pristine h-BN will be relatively immune to the spontaneous reconfiguration of the defect within, perhaps providing a simple pathway to the stabilization of, for example, tailored quantum emitters. This study also raises the importance of studying nanopores stabilities in other two-dimensional materials for advancing both fundamental understanding and applications. Moreover, the reaction at the edge sites of BN nanopores in the air suggests a useful direction to further investigate the reaction mechanism with different gas molecules.

## Methods

**Monolayer hBN synthesis**. The monolayer hBN is synthesized on a copper foil using a CVD process. The quartz tube in the CVD system is first heated to 1070 °C under a flow of 10 sccm of hydrogen ($H_2$) gas. The reaction chamber is maintained in this condition for 10 min to stabilize the temperature. Next, borazine and hydrogen are flowed into the reaction chamber at 0.6 sccm and 10 sccm, respectively, to grow the monolayer hBN. To ensure high-quality monolayer hBN, the furnace is cooled down quickly to room temperature.

**AB stacked multilayer hBN synthesis**. The AB stacked hBN used in this work is synthesized on iron (Fe) foil using a low-pressure (LP)-CVD system, which consists of two heating zones. Ammonia-borane power is used as a solid B and N precursor. During the synthesis, the foil is first annealed for 1 h at 1100 °C under a

flow of 100 sccm $H_2$ and 300 sccm argon (Ar). To grow the AB stacked multilayer hBN. Ar gas is turned off, and the $H_2$ gas flow rate is maintained at 100 sccm. Totally, 100 mg of ammonia borane loaded in the separate quartz tube upstream of the reaction tube is heated to 80 °C to initiate the film growth. After 1 h synthesis, the precursor and main reaction tube are both quickly cooled to room temperature. The $H_2$ gas flow rate is reduced to 10 sccm during the cooling process.

**TEM for nanopore formation and imaging**. The nanopores in h-BN are prepared using a JEOL 2010 TEM operated at 80 kV. The vacuum system is operated in the $10^{-9}$ torr range. To form nanopores, the electron beam is condensed to a 10–20 nm diameter at spot size 3, alpha = 3, with a beam current of ~40 A/cm². In a monolayer hBN, it takes around 2-3 minutes (electron dose: $1.5$–$2.2 \times 10^{21}$ electrons/cm²) to form one pore in a newly exposed layer. In multilayer hBN, the nanopore formation time varies according to the membrane thickness. For a 10 nm thick hBN, it could take two to three hours (electron dose: $0.9$–$1.3 \times 10^{23}$ electrons/cm²) to form a pore. To acquire images, the beam is expended to reduce the beam current down to ~3 A/cm² at spot 3.

## Data availability

All data are available in the main text or the supplementary materials.

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

## Acknowledgements

This work was supported primarily by the Director, Office of Science, Office of Basic Energy Sciences, Materials Sciences and Engineering Division, of the US Department of Energy under contract no. DE-AC02-05-CH11231, within the Nanomachines Program (KC1203), which provided for the design of the project, device fabrication, and TEM characterization. This work was also supported in part by the Director, Office of Science, Office of Basic Energy Sciences, Materials Sciences and Engineering Division, of the US Department of Energy under contract no. DE-AC02-05-CH11231, within the van der Waals Heterostructures Program (KCWF16), which provided for the growth of selected hexagonal boron nitride. J.H.P. and J.K. acknowledge the support from the U.S. Army Research Office (ARO) MURI project under grant number W911NF-18-1-04320431 and the US Army Research Office through the Institute for Soldier Nanotechnologies at MIT, under cooperative agreement No. W911NF-18-2-0048.

## Author contributions

C.D. and A.Z. conceived the idea; C.D., J.H.P., K.W., and T.T. synthesized the materials; C.D., D.P., and C.S. conducted TEM experiments; A.Z. and J.K. supervised the project; and all authors contributed to the discussion of the results and writing of the paper.

## Competing interests

The authors declare no competing interests.
