## [Peer Review File · Communications Chemistry]

Evolution of Nanopores in Hexagonal Boron NitrideReviewers' comments:

Reviewer #1 (Remarks to the Author):

The Authors report the results of TEM experiments aimed at assessing the air and vacuum stability of nanopores in single- and multi-layer h-BN samples, which were created using focused electron beam, and show that the pores are stable in vacuum, but not in air. Taking into account the large amount of attention currently being paid to holey 2D materials in the context of molecule translocation/sieving and beam-mediated structure engineering, these results should be interesting to the community. However, I suggest that the Authors address some issues prior to publication (ideally carrying out additional experiments), as detailed below.

First of all, I do not fully understand why the Authors say that their observations (nanopore evolution under ambient conditions) "contrast with general expectations" and that "conventional wisdom is wrong". At least I would not expect that the pores remain the same in air. It is known that nanopores in graphene are unstable and can easily be filled [[dx.doi.org/10.1021/nl300985q](https://doi.org/10.1021/nl300985q)]. Moreover, it has been discussed [<https://www.pnas.org/doi/full/10.1073/pnas.1400767111>] how pore filling can be prevented. The Authors should better explain why the reader should expect different behavior of graphene and h-BN with holes.

As for the findings the Authors report, an interesting point (which is different to what has been observed in graphene) is that the pores are not filled, but seem to increase (or at least change their shape) due to etching of the edge atoms. This gives rise to a question: can the pores be actually filled, e.g., with carbon atoms to make quantum dots, as discussed previously [<https://journals.aps.org/prb/abstract/10.1103/PhysRevB.87.035404>; <https://journals.aps.org/prl/abstract/10.1103/PhysRevLett.107.035501>]?

I suggest (but do not insist, as it can be technically too difficult) that the Authors expose the samples with pores to different gases at room (or possibly elevated) temperature, obtain the TEM images and report their observations. Ideally the Authors should find the conditions when the pore is completely filled or stabilized by impurity atoms (as reported for graphene, see above) or increases. For example, can exposure to hydrogen stabilize the edges? Reporting such findings would make the impact of the manuscript much stronger.

Minor issues:

What is the difference between "undefected" and pristine regions? Does "undefected" mean that defects were present, but disappeared?

Page numbers in Refs.3, 12, 18, 20,21 appear to be wrong.

Reviewer #2 (Remarks to the Author):

The paper by Ch. Dai et al. reports on the stability of nanopores in hexagonal BN and states by conventional TEM analysis that the pores are stable only under vacuum. It is technically carefully prepared and well written. The experiments show the main message very well and the paper is interesting to the 2D materials community. But it is not surprising since TEM publications have already demonstrated the influence of surface contamination on nanopore growth. For this reason and because of the following critical remarks, I cannot recommend the paper for publication in Communication Chemistry. I suggest after improvement, to publish in a more specified 2D materials journal.

- In the introduction, the study is motivated by the application of nanopores in the field of DNA

sequencing and quantum information technology. However, since the authors state that the pores are only stable in vacuum, they should logically state that nanopores in h-BN cannot be used for DNA detection and quantum sensing, since it is arguably not possible to maintain vacuum conditions in the experiments. Or comment on how the derived knowledge affects the applicability.

- The term nanopore defect is confusing to me. Do you mean that the pore itself is a defect? Mostly the term is used in the manuscript without defect, please be consistent or describe which defect of the nanopore you mean. In the introduction, the term defect is used for individual point defects.
- The electron dose for the experiments should be given in order to check the compatibility.
- The vacuum pressures need to be given always.
- The scheme in Fig. 2 is only an estimate since you don't have any HRTEM results. You write: "The schemes are generic representations of the atomic structures of the nanopore defects (and do not necessarily represent a perfect one-to-one atomic mapping to the TEM images below)." Then why are you presenting them?
- The conclusion in the summary paragraph are not always supported by the experiments. It reads like an opinion: "The changes involve both boron and nitrogen removal at the pore edges, as well as contamination adsorption." You do not present any experiment at atomic resolution. And further: "We attribute the mechanism to reactive oxidation chemistries of the dangling bonds at the edge of the nanopore defects." Also here, there is no experimental evidence (and no citation). And further: "This study also provides guidance to the use of nanopore defects in other two-dimensional materials generally for fundamental studies and applications." From my experience this may not be true. Every 2D material in particular the oxygen-sensitive TDMs are rather different in their behaviour at ambient conditions and need to be studied separately in detail.

Reviewer #3 (Remarks to the Author):

This work shows the formation of the holes is affected by the stacking type of hBN and thus it can attract the researchers in the various fields including Electron Microscopy, Materials Science, Chemical Engineering, and Biomaterials. However, this paper needs to be revised due to many uncertain points.

1. Microscopic methodology

In the current version, the authors mentioned 80kV TEM. Because this work gives a kind of methodology to manipulate holes in hBN, the authors should provide the detailed information such as the beam current, aperture size, beam converging parameters, and so on.

2. TEM mode to make holes?

TEM micrographs show the the holes became already too big to reconstruct the hole shape and edges around holes. Why did the authors utilize TEM rather than STEM?

3. First paragraph/page 6

The contamination regions of the samples are usually disappeared in the TEM with the very high vacuum, though contamination species can sometimes make bonding at the edges with dangling bonds. However, I'm not convinced at all by the relevant micrographs that the contamination is more likely growing regardless of edges. It would be better to use another micrographs with the higher magnification.

4. Identification of the stacking type

One of the interesting points is that the formation of holes depends on the stacking type. Therefore, the authors need to provide how the stacking types (mono, AA', AB) can be determined. And please explain how to control them.

Reply to the Reviewers' comments and list of changes made to the revised manuscript:

We thank the Reviewers for the critical review of the manuscript and the very helpful suggestions for improvement. We have implemented those suggestions in a revised manuscript. In the following, we provide our response to the Reviewers' comments and detail the changes made to the manuscript. Original Reviewer comments are in italics; our responses follow in plain text.

Reviewer(s)' Comments to Author:

Reviewer: 1

Comments:

The Authors report the results of TEM experiments aimed at assessing the air and vacuum stability of nanopores in single- and multi-layer h-BN samples, which were created using focused electron beam, and show that the pores are stable in vacuum, but not in air. Taking into account the large amount of attention currently being paid to holey 2D materials in the context of molecule translocation/sieving and beam-mediated structure engineering, these results should be interesting to the community.

Response: We thank the Reviewer for the positive comments.

However, I suggest that the Authors address some issues prior to publication (ideally carrying out additional experiments), as detailed below.

- *First of all, I do not fully understand why the Authors say that their observations (nanopore evolution under ambient conditions) “contrast with general expectations” and that “conventional wisdom is wrong”. At least I would not expect that the pores remain the same in air. It is known that nanopores in graphene are unstable and can easily be filled [[dx.doi.org/10.1021/nl300985q](https://doi.org/10.1021/nl300985q)]. Moreover, it has been discussed [<https://www.pnas.org/doi/full/10.1073/pnas.1400767111>] how pore filling can be prevented. The Authors should better explain why the reader should expect different behavior of graphene and h-BN with holes.*

Response: We thank the Reviewer for providing the graphene references, which we have incorporated into the manuscript. The main reason that most researchers believe h-BN nanopores should be more stable than graphene nanopores is that the sp²-bonds in h-BN are generally more stable than the sp²-bonds of planar carbon. It is well documented and appreciated that bulk h-BN has far greater chemical and thermal resistance than does graphite/graphene, and this resistance extends to nanoscale morphologies (e.g. nanotubes) as well. Many researchers currently exploring nanopores for DNA sequencing have moved on from graphene, based on the

explicit assumption that h-BN nanopores have a superior well-defined geometry and are more stable than their graphene counterparts (see, for example, refs R1, R2, and R3.). Ours is the first work to test if this is actually the case.

List of changes made: Clarified the current understanding of BN nanopore stability in recent application-based works, on p.4. Discussed graphene nanopore stability and added the references suggested by the reviewer, on p.4.

[R1]: Liu, K., Lihter, M., Sarathy, A., Caneva, S., Qiu, H., Deiana, D., ... & Radenovic, A. (2017). Geometrical effect in 2D nanopores. *Nano letters*, 17(7), 4223-4230. (Ref. 15 in the manuscript)

[R2]: Zhou, Z., Hu, Y., Wang, H., Xu, Z., Wang, W., Bai, X., ... & Lu, X. (2013). DNA translocation through hydrophilic nanopore in hexagonal boron nitride. *Scientific reports*, 3(1), 3287. (Ref. 21 in the manuscript)

[R3]: Liu, S., Lu, B., Zhao, Q., Li, J., Gao, T., Chen, Y., ... & Yu, D. (2013). Boron nitride nanopores: highly sensitive DNA single - molecule detectors. *Advanced materials*, 25(33), 4549-4554. (Ref. 22 in the manuscript)

- *As for the findings the Authors report, an interesting point (which is different to what has been observed in graphene) is that the pores are not filled, but seem to increase (or at least change their shape) due to etching of the edge atoms. This gives rise to a question: can the pores be actually filled, e.g., with carbon atoms to make quantum dots, as discussed previously [<https://journals.aps.org/prb/abstract/10.1103/PhysRevB.87.035404>; cc]?*

Response: Yes, the increasing size of the nanopores in h-BN is a key discovery. As to the question if carbon (for example) could fill the holes to form quantum dots, the answer is nuanced. Detailed theoretical work has investigated this question (e.g. the reference cited by the Reviewer), and indeed some experimental work has shown carbon atom substitutions (ref. R4) or even graphene patches in the h-BN fabric (ref. R5), but these in-fabric substitutions do not appear to happen spontaneously under ambient conditions. Hence, such mechanisms are unlikely to play a role under room-temperature conditions.

[R4]: Mendelson, N., Chugh, D., Reimers, J.R., Cheng, T.S., Gottscholl, A., Long, H., Mellor, C.J., Zettl, A., Dyakonov, V., Beton, P.H. and Novikov, S.V.(2021). Identifying carbon as the source of visible single-photon emission from hexagonal boron nitride. *Nature Materials*, 20(3), 321-328.

[R5]: Teng, S., Song, X., Du, H., Nie, Y., Chen, Y., Ji, Q., Sun, J., Yang, Y., Zhang, Y. & Liu, Z. (2015). Temperature-triggered chemical switching growth of in-plane and vertically stacked graphene-boron nitride heterostructures. *Nature Communications*, 6(1) (2015), 6835.

- *I suggest (but do not insist, as it can be technically too difficult) that the Authors expose the samples with pores to different gases at room (or possibly elevated) temperature, obtain the TEM images and report their observations. Ideally the Authors should find the conditions when the pore is completely filled or stabilized by impurity atoms (as reported for graphene, see above) or increases. For example, can exposure to hydrogen stabilize the edges? Reporting such findings would make the impact of the manuscript much stronger.*

Response: We appreciate the suggestion of these extended experiments, but, as the Reviewer surmises, they are technically exceedingly difficult and well beyond the scope of the present study.

- *What is the difference between “undefected” and pristine regions? Does “undefected” mean that defects were present, but disappeared?*

Response: We apologize for the confusing nomenclature. What we meant by “undefected” was the pristine regions, with no nanopores or surface contamination present. We have now provided a clearer and more consistent terminology throughout the manuscript.

List of changes made: Changed, where appropriate, “undefected” to “pristine” throughout the manuscript.

- *Page numbers in Refs.3, 12, 18, 20,21 appear to be wrong.*

Response: Again, our apologies for these typos.

List of changes made: Updated the page number for Refs. 3, 12, 18, 20, 21 in the reference list.

Reviewer #2:

The paper by Ch. Dai et al. reports on the stability of nanopores in hexagonal BN and states by conventional TEM analysis that the pores are stable only under vacuum. It is technically carefully prepared and well written. The experiments show the main message very well and the paper is interesting to the 2D materials community.

Response: We thank the Reviewer for the positive characterization.

But it is not surprising since TEM publications have already demonstrated the influence of surface contamination on nanopore growth. For this reason and because of the following critical remarks, I cannot recommend the paper for publication in Communication Chemistry. I suggest after improvement, to publish in a more specified 2D materials journal.

Response: We are unaware of any previous work (published or unpublished) that has examined individual h-BN nanopore evolution in vacuum or air over extended time periods.

• *In the introduction, the study is motivated by the application of nanopores in the field of DNA sequencing and quantum information technology. However, since the authors state that the pores are only stable in vacuum, they should logically state that nanopores in h-BN cannot be used for DNA detection and quantum sensing, since it is arguably not possible to maintain vacuum conditions in the experiments. Or comment on how the derived knowledge affects the applicability.*

Response: Yes, that is exactly our point, and we agree it should be emphasized.

List of changes made: We have added text to the introduction to make it more explicit that as-created h-BN nanopores are unlikely to serve well in applications that require atomically precise edge structure of the nanopores.

• *The term nanopore defect is confusing to me. Do you mean that the pore itself is a defect? Mostly the term is used in the manuscript without defect, please be consistent or describe which defect of the nanopore you mean. In the introduction, the term defect is used for individual point defects.*

Response: We very much appreciate this comment. Yes, we meant that the pore itself could be termed a defect. But we now realize that the terminology in our original manuscript was very confusing and somewhat inconsistent (it also doesn't help that different authors use different terms to describe nanopores in 2D materials, including calling them defects). We have now simplified and unified our terminology in the revised manuscript. We no longer describe nanopores as defects. If the material is multi-layered and the hole or channel doesn't go through all layers, we term it a pit or "incomplete nanopore".

List of changes made: Simplified and unified nanopore terminology throughout manuscript. The term nanopore defect has been eliminated, and if the term defect is used, it is made clear what is meant.

• *The electron dose for the experiments should be given in order to check the compatibility.*
• *The vacuum pressures need to be given always.*

Response: Agreed.

List of changes made: We have added information about vacuum level and required doses as "Supplementary Nanopore Formation and Imaging", on p.1 of Supplementary Information. The dose required to form the nanopore in a monolayer hBN is around $1.5- 2.2 \times 10^{21}$ electrons/cm². For multilayer hBN, the required doses vary according to the film thickness. A 10nm thick hBN requires an electron dose of $0.9- 1.3 \times 10^{23}$ /cm². The JOEL 2010 TEM is operated at the vacuum range of 10^{-9} torr.

• *The scheme in Fig. 2 is only an estimate since you don't have any HRTEM results. You write: "The schemes are generic representations of the atomic structures of the nanopore defects (and do not necessarily represent a perfect one-to-one atomic mapping to the TEM images below)." Then why are you presenting them?*

Response: We believe the schemes as presented are instructive, despite the possible "scale" differences. For example, if a very small triangular nanopore can readily be incorporated into h-BN (which it cannot in graphene), then by extension a much larger triangular nanopore, with the same orientation, can also be incorporated in h-BN.

List of changes made: Clarified the caption for schemes in Figure 2, p.14.

• *The conclusion in the summary paragraph are not always supported by the experiments. It reads like an opinion: “The changes involve both boron and nitrogen removal at the pore edges, as well as contamination adsorption.” You do not present any experiment at atomic resolution. And further: “We attribute the mechanism to reactive oxidation chemistries of the dangling bonds at the edge of the nanopore defects.” Also here, there is no experimental evidence (and no citation). And further: “This study also provides guidance to the use of nanopore defects in other two-dimensional materials generally for fundamental studies and applications.” From my experience this may not be true. Every 2D material in particular the oxygen-sensitive TDMs are rather different in their behaviour at ambient conditions and need to be studied separately in detail.*

Response: We agree with the Reviewer that different 2D materials will undoubtedly display differences in nanopore evolution, and therefore not all details pertaining to nanopore evolution in h-BN is applicable to all other materials. But we do feel that many of our observations have relevance to other 2D materials, and researchers should be informed about that, even if they are not working on h-BN. We admit that our study did not explicitly investigate the detailed chemistry of nanopore evolution in h-BN, but we did not make that claim. We also feel it can be useful to present a hypothesis or speculate on possible nanopore evolution mechanisms, as such speculations may motivate or guide further theoretical or experimental study. Nevertheless, to avoid the perception of overselling our findings, we have made changes to our presentation with respect to mechanisms and relevance to other 2D materials.

List of changes made: Updated the conclusion, p.8, based on the Reviewers’ suggestions: 1) deleted the proposed reaction mechanism at the dangling bond, and 2) updated the comments about the impact for other 2D materials.

Reviewer #3:

This work shows the formation of the holes is affected by the staking type of hBN and thus it can attract the researchers in the various fields including Electron Microscopy, Materials Science, Chemical Engineering, and Biomaterials.

Response: We thank the Reviewer for the positive comments.

However, this paper needs to be revised due to many uncertain points.

- *Microscopic methodology. In the current version, the authors mentioned 80kV TEM. Because this work gives a kind of methodology to manipulate holes in hBN, the authors should provide the detailed information such as the beam current, aperture size, beam converging parameters, and so on.*

Response: We agree.

List of changes made: Updated the details of TEM operation condition and information on p.1 of supplementary information. In this work, the nanopores in h-BN are prepared using a JEOL 2010 TEM operated at 80 kV. Vacuum system is operated in the 10^{-9} torr range. To form nanopores, the electron beam is condensed to a 10–20 nm diameter at spot size 3, alpha=3 with a beam current of ~ 40 A/cm². The doses used for forming the nanopores in monolayer or multilayer hBN are $1.5- 2.2 \times 10^{21}$ electrons/cm² and $0.9- 1.3 \times 10^{23}$ electrons/cm², respectively. To acquire images, the beam is expended to reduce the beam current down to ~ 3 A/cm² at spot 3. We now provided this information in the supplementary information.

- ***TEM mode to make holes? TEM micrographs show the the holes became already too big to reconstruct the hole shape and edges around holes. Why did the authors utilize TEM rather than STEM?***

Response: In this work, we are focusing on understanding the evolution of nanopores/defects caused by the environment, in vacuum or in air, rather than a detailed study of electron beam irradiation. Therefore, in our evolution (time) imaging sequences, we strive to reduce the effect of electron beam during imaging process. In STEM, the acquisition time for imaging is longer than for TEM. Also, the electron beam scanning time varies from pixel to pixel in STEM. Therefore, in STEM it is often difficult to distinguish changes in sample geometry due to electron irradiation, from those that have originated from non-electron-beam chemical processes.

List of changes made: Motivation for using TEM rather than STEM is added in the manuscript, on p. 4.

- ***First paragraph/page 6. The contamination regions of the samples are usually disappeared in the TEM with the very high vacuum, though contamination species can sometimes make bonding at the edges with dangling bonds. However, I'm not convinced at all by the relevant micrographs that the contamination is more likely growing regardless of edges. It would be better to use another micrographs with the higher magnification.***

Response: These helpful comments from the Reviewer point to a subtlety in h-BN nanopore evolution that was perhaps not presented as clearly as it should have been in the original manuscript. When nanopores are created in h-BN via electron irradiation, one typically ends up with a nanopore (or nanopores) within relatively clean patches of h-BN. However, nearby there can be regions of rather severe surface contamination. Our study shows that in vacuum, the nanopores in the clean region do not change, and the nearby contamination also does not change or expand. However, in air, two things can happen. First, the nanopores themselves evolve (typically growing in size), and the regions of nearby contamination can also grow or expand. The nucleation sites for this contamination “spreading” are typically existing patches of contamination. It is possible that the expanding or spreading of this surface contamination obscures (or overflows) existing nanopores, thus rendering them inoperative. To make these points more clear, we have interchanged the order of Figs. 3 and 4, and have modified and

streamlined our discussion of these figures. New Fig. 4 also has better graphics to show the surface contamination spreading in the air-exposed sample

List of changes made: Switched Figs. 3 and 4, and updated discussion in text thereof. Updated graphics on new Figs. 4b,d to highlight surface contamination spreading and obscuring of some nanopores.

- **Identification of the stacking type. One of the interesting points is that the formation of holes depends on the stacking type. Therefore, the authors need to provide how the stacking types (mono, AA', AB) can be determined. And please explain how to control them.**

Response: Good points. Our group's earlier work (DOI: 10.1088/0953-8984/25/4/045009) presented a detailed study on h-BN stacking orders. The widely used h-BN crystals grown by Kenji Watanabe and Takashi Taniguchi are always AA' stacked. As stated in our earlier work, we employed LP-CVD on a transition metal surface to dictate the stacking growth sequence of h-BN. The conventional AA' growth could be changed exclusively to Bernal AB growth on both Cu and Fe substrates. The stacking order could be identified by the SAED pattern of a thick region of multilayer. The ratio of the intensities of the first order $\langle 0\ 1\ 0 \rangle$ and second order $\langle 1\ 1\ 0 \rangle$ peaks are ~ 0.3 , which suggest that the tacking is AB.

List of changes made: The information on how to select and identify the stacking order is clarified in the manuscript, on p.5. Also, we explicitly refer to our previous work for further details.

REVIEWERS' COMMENTS:

Reviewer #1 (Remarks to the Author):

I am satisfied with the response of the Authors to my comments and comments of other reviewers and the revisions they made. I recommend the manuscript for publication in its present form.

Reviewer #3 (Remarks to the Author):

I do appreciate the authors' efforts to reflect every single comment. The revised manuscript makes me more clear about all results in this work.

Reply to the Reviewers' comments and list of changes made to the revised manuscript:

Reviewer(s)' Comments to Author:

Reviewer #1

Comments:

I am satisfied with the response of the Authoes to my comments and comments of other reviewers and the revisions they made. I recommend the manuscript for publication in its present form.

Response: We appreciate the Reviewer's suggestion of acceptance for publication.

Reviewer #3

Comments:

I do appreciate the authors' efforts to reflect every single comment. The revised manuscript makes me more clear about all results in this work.

Response: We thank the Reviewer for the positive comments.